# Personal Data Exchange and Agentic AI governance with Consent Receipts

Mark Lizar

**Abstract**

This submission advocates for the implementation of consent receipts to authorize the exchange of personal data, aligning with the principles of transparency and trust in data governance. It builds on the Glassbox Data Governance model, which emphasizes secure transparency and co-regulation of identification management systems. The use of consent receipts, as outlined in ISO/IEC 27560:TS:2023, provides a standardized framework for notice and consent record information, ensuring open and free access.

## 1. Introduction

The Open Consent Group, through initiatives like the Kantara Consent Receipt, has been instrumental in developing standards for digital consent and data sovereignty. The evolution of transparency standards necessitates a shift from analogue privacy procedures to digital notice and consent records. This is supported by international frameworks such as the OECD's guidance on enhancing access to and sharing of data in the age of artificial intelligence.

## 2. Glassbox Data Governance and Consent Receipts

Glassbox Data Governance proposes a technical architecture for regulated international transparency and consent standards in data processing. It embeds secure transparency to minimize exposure and unregulated sharing of personal data, enhancing privacy and security by default. The use of consent receipts, akin to transaction receipts, provides individuals with a record of their digital identity relationships. This enables them to manage their data access and use dynamically, similar to how receipts are used in banking to track transactions.

## 3. Implementing Notice Records and Consent Receipts for Personal Data Exchange, for real consent based data control

Consent receipts are crucial for addressing common problems such as unread policies and lack of tracking capabilities. They promote user awareness and provide a mechanism for individuals to withdraw consent easily. By implementing a transparency-centric governance model, consent receipts can enhance trust in data flows and mitigate privacy risks associated with digital identity management.

## 4. Key Recommendations

1. **Standardization of Consent Receipts:** Implement standardized consent receipts as a digital notice record, enabling individuals to track and manage their personal data exchanges securely.
2. **Legislative Enforcement:** Align legislative frameworks with international standards for data governance, ensuring that consent receipts are recognized as a lawful basis for data processing.
3. **Interoperability:** Ensure that consent receipts are interoperable across different legal jurisdictions and digital identification technologies to facilitate seamless data exchange.

---

*3rd Privacy & Personal Data Management Session, colocated with Solid Symposium 2025, April 24–25, 2025, Leiden, Netherlands*

4. **Transparency and Trust:** Enhance transparency in data access and sharing arrangements to encourage responsible data governance practices, fostering trust in the data ecosystem.

## 5. Conclusion

The adoption of consent receipts as a mechanism for authorizing personal data exchange offers a robust solution for enhancing transparency and trust in data governance. By leveraging international standards such as ISO/IEC 27560 and aligning with legislative frameworks, we can create a more secure and transparent environment for personal data management. This approach not only supports individual rights but also promotes a more accountable and trustworthy data ecosystem.

This submission underscores the importance of consent receipts in modernizing data governance and ensuring that personal data exchanges are secure, transparent, and trustworthy. By integrating these receipts into legislative and technological frameworks, we can significantly enhance individual control over personal data and foster a more robust data ecosystem.

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

## A. Conditions for Consent

- **Recommendations for Section 40A:** Enhance consent conditions by requiring a record and receipt for assurance, aligning with digital privacy scopes that necessitate transparency in data processing.

## B. Implementation Framework

- **Controller Identification and Consent Receipts:** Mandate controllers to provide identification and a notice record for online services, ensuring individuals receive a consent receipt for personal data processing.
- **Gateway Signaling:** Utilize government gateway consent receipts to facilitate secure attribute verification and validation, maintaining anonymity while enabling government services.