# OpenReview forum: "Personal Data Exchange and Agentic AI governance with Consent Receipts"
_SolidProject.org/SoSy/2025/Privacy_Session — Submitted to SoSy2025-Privacy_

### Official Review · ~Ines_Akaichi1 · 2025-03-25
**The idea  of incorporating consent receipts in personal data exchang is interesting but the abstract is too high level.**

**Rating:** 5
**Confidence:** 4

**Review:**

This abstract introduces the idea of incorporating consent receipts in personal data exchanges. While the idea is intriguing, it remains at a high level and lacks sufficient detail. Below are questions and suggestions for improving the abstract.

It is unclear how consent receipts would address unread policies. While their role in tracking consent is somewhat evident, their specific function in this context needs further clarification.
Who exactly benefits from consent receipts? The author compares them to transaction receipts in banking. Does this imply that once a user consents to using their data, they receive a consent receipt? If so, what information does this receipt contain? Is it something users can access on demand? These questions also tie back to the initial concern: how do consent receipts relate to unread policies? How are these consent receipts used for tracking data?

Overall, I agree with the key recommendations. However, the abstract would benefit from a figure illustrating the use of consent receipts in data exchange because this is not clear to me. This can depict the involved entities and a high-level process outlining the necessity of consent receipts.

On a side note; make sure you include references inside the text if you are presenting the references at the end.

---

### Official Review · ~Julian_Flake1 · 2025-03-27
**Good idea, but lack of context, innovation and contribution.**

**Rating:** 3
**Confidence:** 3

**Review:**

The submissions "Personal Data Exchange and Agentic AI governance with
Consent Receipts" describes the idea to use consent receipts, a term
and concept initially defined by the Kantara initiative and later
standardized in ISO/IEC 27560:TS:2023, to represent users' decisions
with regard to the processing of their personal data.

The challenge of collecting, managing and representing consent is
crucial for establishing trustworthy and legally compliant systems
that process personal data. It seems that issues around consent
management are not yet fully solved in the Solid project. The topic is
especially challenging due to the decentralized design of Solid's
architecture, since actors have to regularly deal with different
requirements from different jurisdictions, which also create define
different conditions related to gathering and managing consent.

Unfortunately, the submission does not extend further beyond
describing an idea (use standardized forms and formats) in the context
of the Solid project, which is a good, but maybe not an exclusively
new idea. While it's worth to discuss this topic, the submission does
not provide enough context, specific suggestions or new insights to
stimulate an insightful discussion, in my opinion.

Suggestions for speficic improvements
- Describe the current state on the topic of consent in the Solid
  project (e.g. "consider
  https://www.w3.org/TR/privacy-principles/#consent-principles")
- The topic of consent management is broad. Existing and related
  solutions are not described.
- The title mentions "agentic AI governance". I'm not an AI expert
  (maybe just like other participants of the symposium) and I   don't
  know that concept. The text does not further refer to the   AI
  aspect, except for mentioning the existence of an OECD report
  about accessing and sharing of data in the AI era in the
  introduction. No further details of that report about consent
  management and the potential relationship with Solid are
  described.
- The submission mentions "the Glassbox Data Governance Model"
  without providing reference. For me, it's not clear, whether the
  author introduces a new idea/term here or if he refers to some
  existing model. I think the latter is the case, but I can't find
  any reference.
- Use your references in the text.

---

### Decision · Program_Chairs · 2025-04-01

Reject